# The Impact of Reduced Towing Fees on Vehicle Redemption Rates for Low-Income Individuals: Evidence from San Francisco's 2020 Policy

**Google Gemini**    Min Min Fong[*]    Abhishek Nagaraj[†]

## Abstract

This paper examines the impact of an August 2020 San Francisco policy that drastically lowered towing fees for low-income individuals. Leveraging a comprehensive dataset of towing incidents, we employ a difference-in-differences design to estimate the causal effect of the fee reduction on vehicle redemption rates. Our analysis shows the policy was highly effective, increasing the redemption rate for a low-income proxy group, with the effect being concentrated among "poverty tows" (e.g., for unpaid tickets) and was largest for owners of the oldest vehicles, suggesting the financial relief was most impactful where it was most needed. Our results provide rigorous evidence that direct financial relief can prevent asset forfeiture and promote equity in municipal fine enforcement.

## 1 Introduction

In dense urban environments, vehicle towing is a primary tool for managing public space, yet its implementation often creates a sharp conflict between municipal goals of order and the financial stability of residents. This tension is particularly acute in San Francisco, where some of the nation's highest towing fees can transform a parking infraction into a significant economic shock. For low-income individuals, the cost of retrieving a vehicle can be insurmountable, leading to the loss of a critical asset and cascading consequences for employment and housing. These harms are most pronounced in the case of "poverty tows"—tows initiated not for immediate safety hazards, but for administrative non-compliance such as unpaid parking tickets or expired registration, which disproportionately affect those with limited financial means.

This paper evaluates the causal impact of San Francisco's August 2020 tow fee waiver, a reform that substantially reduced towing and storage fees for eligible low-income individuals. The waiver represents a direct financial intervention, designed to address the regressive burden of towing by making vehicle retrieval attainable for vulnerable populations. A subsequent June 2021 policy change altered the conditions under which certain debt-related tows could occur, primarily protecting unhoused individuals from being towed for unpaid citations. While important, this latter reform serves here primarily as a robustness check, as it had little measurable effect on redemption rates.

Using a comprehensive administrative dataset of all towing incidents, this study employs a difference-in-differences (DiD) framework to isolate the causal effects of these policies. Because individual-level income data are unavailable, we follow established practice in transportation and economic research by constructing a proxy for owner socioeconomic status based on vehicle age and make [e.g., 9, 20].

Our findings demonstrate that the August 2020 fee waiver was a highly effective intervention, causing a 3.5 percentage point increase in the vehicle redemption rate for the low-income proxy group. This effect was concentrated among "poverty tows" and was largest for the owners of the oldest

---
[*]Haas School of Business, University of California, Berkeley. Email: zmfong@berkeley.edu
[†]Haas School of Business, University of California, Berkeley. Email: nagaraj@berkeley.edu

vehicles, suggesting the financial relief was most impactful where it was most needed. In contrast, the 2021 rule change had no statistically significant effect on redemption rates, underscoring that procedural adjustments alone are insufficient. Taken together, these results provide robust evidence that direct financial relief can be a powerful tool in preventing asset forfeiture and mitigating the regressive impacts of municipal fine and fee enforcement.

## 2 Literature review

### 2.1 The Punitive Landscape of Municipal Fines and Vehicle Towing

From an economic perspective, municipal fines are a tool of optimal deterrence. The foundational model of crime and punishment [1] posits that a rational individual commits an offense only if the expected utility of the act exceeds its expected cost, which is a function of the probability of apprehension and the severity of the sanction. Within this framework, fines are considered the most efficient form of punishment because they are a simple transfer payment from the offender to the state, imposing minimal deadweight loss on society [1]. The state, acting as a benevolent social planner, should therefore rely on fines to force offenders to internalize the external costs of their actions, reserving more socially costly sanctions like incarceration for when the optimal penalty exceeds an individual's ability to pay [14, 17].

However, the theoretical efficiency of this model breaks down when confronted with real-world wealth inequality and credit constraints. A core assumption is that a fixed fine represents a consistent punishment, but this is violated by the diminishing marginal utility of income. A flat-rate fine is inherently regressive, imposing a trivial utility loss on a high-income individual while representing a catastrophic burden for a person with low income [4, 6]. When credit constraints bind, preventing low-income individuals from borrowing to pay a large, unexpected fine, the sanction ceases to be an efficient transfer and instead triggers cascading consequences such as driver's license suspensions that sever access to labor markets and even incarceration for non-payment [6, 5].

A growing body of public finance literature offers a positive, rather than normative, model for this behavior, positing that municipalities may not act as social welfare maximizers but as revenue-maximizing agents. This "taxation by citation" hypothesis suggests that cities systematically use their enforcement powers to generate revenue, particularly in response to fiscal stress [3, 12]. Empirical studies have found a strong link between economic downturns, such as the Great Recession, and a subsequent increase in municipal reliance on revenue from fines and fees [3]. This can distort law enforcement priorities away from public safety and toward easily citable, revenue-positive offenses [18].

Vehicle towing for non-safety violations is a prime example of these economic tensions. The seizure of a vehicle for unpaid tickets or expired registration functions as an aggressive debt collection tool, not a public safety measure [10]. The 2019 report *Towed Into Debt* found that such "poverty tows" are far more likely to result in the permanent loss of the vehicle. In San Francisco, 50% of vehicles towed for unpaid tickets and 57% of those towed for lapsed registration were sold at auction, compared to a citywide average of just 9% [19]. This stark divergence highlights how financial precarity drives asset forfeiture. The system has also faced legal challenges, culminating in a 2023 California Court of Appeal ruling that found San Francisco's policy of towing legally and safely parked vehicles solely for unpaid parking tickets to be an unconstitutional seizure [2].

### 2.2 A Behavioral Framework for Decision-Making Under Financial Constraint

The descriptive literature documents a system that is demonstrably harmful to low-income individuals. To understand why this population is so vulnerable, we turn to behavioral economics, which offers a framework for how financial constraints fundamentally alter cognition and decision-making.

Classical economic theory's model of a perfectly rational actor is challenged by behavioral economics, which integrates insights from psychology to build a more realistic model of human choice based on bounded rationality [16]. This is often explained through a two-system model of the brain, where the deliberate, effortful "System 2" can be depleted by stress and cognitive overload, leaving individuals more reliant on the error-prone "System 1" [7].

In their seminal work, *Scarcity: Why Having Too Little Means So Much*, behavioral economist Sendhil Mullainathan and psychologist Eldar Shafir argue that scarcity reshapes cognition [13]. The constant mental work required to manage scarce resources taxes a finite resource they call cognitive bandwidth. Studies show that simply asking low-income individuals to contemplate a major expense can temporarily reduce IQ test performance by up to 13 points, comparable to losing a full night's sleep [11].

This intense focus on immediate problems also leads to "tunneling," where the mind concentrates only on the urgent matter inside the tunnel while other important but less immediate concerns—like a parking ticket—are neglected. Scarcity exacerbates present bias, the tendency to overweight near-term payoffs [8]. For someone under financial duress, the immediate crisis of paying for rent or food dominates, while the future consequence of an unpaid ticket is heavily discounted. This reframes nonpayment as not irresponsibility, but a constrained, adaptive response to immediate priorities.

### 2.3 Gaps in the Literature and This Paper's Contribution

The two bodies of literature reviewed here, while complementary, exist on largely parallel tracks. The primary literature provides a powerful account of the *what* of the towing problem—a punitive system with devastating, racially disparate impacts. The secondary literature offers a compelling theory of the *why*—the cognitive mechanisms of scarcity that explain decision-making under financial duress.

The gap lies between these domains. Few studies empirically test the causal links between the specific enforcement practices described in the primary literature and the cognitive mechanisms elucidated in the behavioral literature. While descriptive work (e.g., [19]) identifies the problem, it does not test the effectiveness of policy reforms. Conversely, behavioral research provides a strong theory for why a fee waiver should work (by reducing immediate financial pressure and freeing up bandwidth), but there is little quasi-experimental evidence in the context of municipal fines.

This paper begins to bridge this gap. Using a large administrative dataset and a difference-in-differences framework, we provide a rigorous evaluation of a real-world policy intervention—a large-scale fee waiver program—designed to alleviate the financial scarcity at the heart of the towing problem. Our study is one of the first to causally estimate the effectiveness of such a policy in changing redemption behavior. Furthermore, by using vehicle characteristics as a proxy for income, we test a key prediction of scarcity theory: that effects will be strongest for those under the greatest constraint (i.e., owners of the oldest vehicles). This provides empirical evidence that connects the *what* of policy with the *why* of behavioral science.

## 3 Setting and data

### 3.1 The San Francisco towing problem and the 2020 policy change

Vehicle towing in San Francisco is a critical but contentious tool for municipal management. In a city characterized by high population density and intense competition for limited curb space, the San Francisco Municipal Transportation Agency (SFMTA) frames its towing program as essential for maintaining public safety, managing traffic congestion, and ensuring streets remain clean and functional. With approximately 42,000 vehicles towed annually, the program is a significant and constant feature of the city's landscape.

However, the high cost of retrieving a towed vehicle—often exceeding $500 and escalating rapidly with daily storage fees—has made San Francisco's towing fees among the highest in the nation. This has created a significant economic burden, particularly for low-income residents, for whom the loss of a vehicle can cascade into the loss of employment, housing, and access to essential services.

A central issue in this context is the practice of "poverty tows"—a term used by advocates to describe tows initiated not for immediate public safety reasons, but for administrative non-compliance strongly correlated with financial hardship, such as tows for having five or more unpaid parking tickets, registration expired for more than six months, or parking in the same spot for more than 72 hours [19]. Data has shown that vehicles towed for these reasons are far more likely to be abandoned and sold at lien sale, suggesting the practice functions less as a corrective measure and more as a confiscatory one for those unable to pay.

In response to these challenges, the city has implemented a series of reforms. This analysis focuses on two distinct and pivotal policy changes.

**The August 2020 Fee Waiver** was a financial intervention spurred by the city's Financial Justice Project and the economic crisis of the COVID-19 pandemic. The SFMTA Board of Directors approved a new budget in April 2020 that included a major financial relief package for towed vehicle owners. This policy, which became operative on August 1, 2020, did not change the reasons a car could be towed but dramatically lowered the cost of retrieval for vulnerable residents [15].

- **For Low-Income Individuals:** The policy established a deep discount, reducing the tow fee to approximately $100 and waiving the administrative fee (which could be over $300). It also provided a waiver for up to 15 days of storage fees.

- **For Individuals Experiencing Homelessness:** The policy created a new, one-time waiver for all tow and administrative fees, making the cost of retrieval $0, and extended the storage fee waiver to 30 days.

As a secondary point of comparison, **the June 2021 Rule Change** introduced procedural limits on debt-related tows. Enacted in response to litigation filed by the Coalition on Homelessness, the new rules restricted towing legally parked vehicles solely for unpaid citations. These rules went into effect on June 21, 2021 [15].

- **Protections for People Experiencing Homelessness:** Prohibited the booting or towing of a vehicle belonging to a verified homeless person solely for having late, unpaid parking citations.

- **New Rules for Debt-Related Tows:** For other individuals, a vehicle with $2,500 or less in late tickets could not be towed without first giving the owner a 72-hour warning to resolve the debt.

Taken together, these reforms represent two distinct interventions. The 2020 policy was a direct financial treatment that reduced the cost of a tow, while the 2021 policy was a procedural adjustment that changed the conditions under which certain tows could occur. This paper focuses primarily on the 2020 fee waiver as the main source of identification, with the 2021 rule change serving as a robustness check.

### 3.2 Data creation and summary statistics

This study utilizes a comprehensive administrative dataset of individual towing incidents provided by the San Francisco Municipal Transportation Agency (SFMTA), covering the period from 2018 through 2025. To evaluate the impact of the city's policy changes, several key variables were constructed.

The primary outcome variable, Redemption, is a binary indicator coded as 1 if a vehicle was retrieved by its owner and 0 if it was ultimately sold at a lien sale or otherwise not reclaimed.

As individual-level income data is not available in the dataset, a Low-Income Proxy variable was created to serve as the primary treatment group. For this analysis, a vehicle is classified into the low-income group if it is a non-luxury make and has a vehicle age of more than 10 years at the time of the tow. All other vehicles serve as the control group.

To analyze the specific nature of the tow, a binary Hardship Tow variable was created. A tow is classified as a "poverty tow" if the violation is one of three non-safety, administrative reasons strongly associated with financial poverty: having five or more unpaid parking tickets, registration expired for more than six months, or parking in the same location for more than 72 consecutive hours.

Finally, a Repeat Tow variable was generated to indicate whether a specific vehicle had been towed previously within the dataset.

Table 1: Summary Statistics by Group and Policy Period

| Characteristic | Control | | Low-Income | |
|---|---|---|---|---|
| | Pre-Policy | Post-Policy | Pre-Policy | Post-Policy |
| Redemption Rate (Mean) | 0.869 | 0.887 | 0.630 | 0.686 |
| Vehicle Age (Mean) | 5.87 | 7.00 | 17.16 | 18.17 |
| Vehicle Age (Median) | 4.00 | 6.00 | 16.00 | 17.00 |
| Poverty Tow (%) | 10.94 | 11.09 | 30.32 | 28.04 |
| Repeat Tow (%) | 27.17 | 27.04 | 33.21 | 37.17 |
| Out-of-State (%) | 11.81 | 11.83 | 12.93 | 14.66 |
| Number of Tows (N) | 75,536 | 106,547 | 32,455 | 55,073 |

Table 1 presents the summary statistics for the low-income and control groups, comparing the periods before and after the implementation of the August 2020 fee waiver policy. The table reveals the significant baseline disparities that existed prior to the policy interventions. In the pre-policy period, the redemption rate for the low-income group was substantially lower than the control group (63.0% vs. 86.9%). The vehicle-based proxy is validated by the stark difference in average vehicle age, with the low-income group's vehicles being nearly three times older (17.16 years vs. 5.87 years).

Furthermore, the data confirms that low-income vehicle owners were disproportionately affected by poverty-related tows, which accounted for 30.3% of their tows compared to just 10.9% for the control group. Following the policy's implementation, the raw data shows a notable increase in the redemption rate for the low-income group, rising from 63.0% to 68.6%. These baseline differences and raw changes provide the foundation for the difference-in-differences analysis used to isolate the causal impact of the policy reforms.

## 4 Results

To estimate the causal impact of San Francisco's towing policy reforms on vehicle redemption rates, we employ a difference-in-differences (DiD) framework. This quasi-experimental approach allows us to compare the change in outcomes for a treated group against a control group, before and after the policy interventions, thereby controlling for unobserved time-varying factors.

Our main specification is as follows:

$$
\begin{aligned}
Y_{it} = {} & \beta_0 + \beta_1 \text{LowIncome}_i + \beta_2 \text{Post2020}_t + \beta_3 (\text{Post2020}_t \times \text{LowIncome}_i) \\
& + \beta_4 \text{Post2021}_t + \beta_5 (\text{Post2021}_t \times \text{LowIncome}_i) \\
& + \gamma_1 \text{PovertyTow}_i + \gamma_2 \text{RepeatTow}_i + \gamma_3 \text{OutOfState}_i \\
& + \delta_m + \theta_d + \epsilon_{it}
\end{aligned}
\tag{1}
$$

where $Y_{it}$ is a binary variable equal to 1 if vehicle $i$ towed at time $t$ was redeemed. $\text{LowIncome}_i$ is an indicator for vehicles in our low-income proxy group (non-luxury makes older than 10 years). $\text{Post2020}_t$ and $\text{Post2021}_t$ are indicators for the periods after the August 2020 fee waiver and the June 2021 rule change, respectively. The coefficients of interest, $\beta_3$ and $\beta_5$, are the DiD estimators for each policy. The model also includes controls for whether the tow was classified as a poverty tow, whether the vehicle was a repeat tow, and whether the vehicle was out of state. Finally, $\delta_m$ and $\theta_d$ denote month–year and day-of-month fixed effects, respectively, which flexibly control for seasonality and other time trends.

To investigate treatment effect heterogeneity, we expand this model into a triple-difference (DDD) specification, interacting the policy variables with more granular subgroups of the low-income proxy

based on vehicle age (k):

$$Y_{it} = \alpha + \sum_k \gamma_k \text{LowIncGroup}_i^k + \delta_1 \text{Post2020}_t$$
$$+ \sum_k \beta_k \left( \text{Post2020}_t \times \text{LowIncGroup}_i^k \right) + \delta_2 \text{Post2021}_t$$
$$+ \sum_k \lambda_k \left( \text{Post2021}_t \times \text{LowIncGroup}_i^k \right) + \delta_m + \theta_d + \epsilon_{it} \qquad (2)$$

In this specification, $\text{LowIncGroup}_i^k$ is a set of indicators for vehicles in age brackets $k \in \{10\text{--}15, 16\text{--}20, > 20\}$. The coefficients of interest, $\beta_k$, estimate the distinct causal effect of the 2020 fee waiver for each age group.

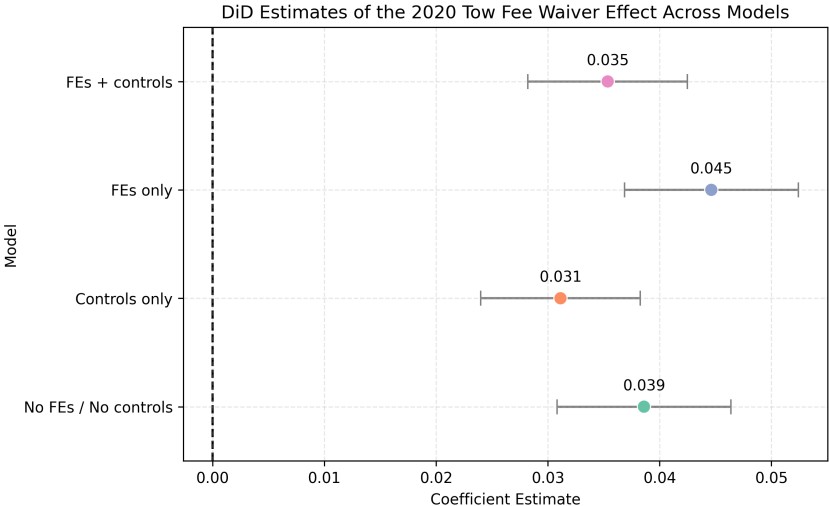

Figure 1: Coefficient Estimates: Robustness to Controls and Fixed Effects

Figure 1 shows that the August 2020 fee waiver produced a clear and statistically significant increase in redemption rates for low-income vehicle owners. In our most saturated model, the interaction term $\text{Post2020}_t \times \text{LowIncome}_i$ is positive and highly significant, indicating that the waiver raised redemption rates by about 3.5 percentage points relative to the control group. This provides strong evidence that the financial relief directly enabled more low-income owners to retrieve their vehicles.

By contrast, the June 2021 rule change shows no measurable effect on redemption. The coefficient on $\text{Post2021}_t \times \text{LowIncome}_i$ is small and statistically insignificant. This is a logical finding, as the 2021 policy was primarily designed to prevent certain tows from occurring in the first place, rather than changing the outcome for vehicles that were still towed for other reasons.

Finally, the baseline disparities are stark: before the reforms, low-income vehicles were about 17.9 percentage points less likely to be redeemed compared to higher-income proxies. This gap underscores the inequities the policies were meant to address.

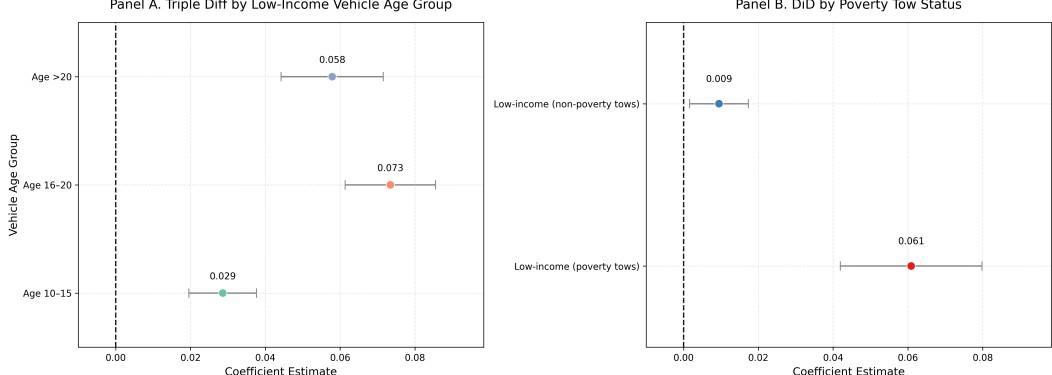

Figure 2: Coefficient Estimates. (a) By low income group. (b) By Poverty Tow Status.

To investigate which segments of the low-income population were most affected by the policies, we turn to the triple-difference model. The results, presented in Panel A of Figure 2, reveal significant heterogeneity in the policy's impact, demonstrating that the benefits of the August 2020 fee waiver were most pronounced for the owners of the oldest vehicles.

Panel A of Figure 2 highlights strong heterogeneity: the 2020 waiver was most impactful for the oldest vehicles. The "dose-response" pattern is striking:

- For cars **10-15 years old**, the policy increased the redemption rate by 2.9 percentage points.
- For cars **16-20 years old**, the policy increased the redemption rate by 7.3 percentage points.
- For cars **over 20 years old**, the policy increased the redemption rate by 5.8 percentage points.

The effect more than doubles for cars older than 15 years compared to younger ones, suggesting that financial relief was most valuable to the most resource-constrained households. In contrast, the 2021 procedural rule change generated only modest and inconsistent effects across age groups, reinforcing that the main driver of redemption gains was the 2020 fee reduction.

Panel B of Figure 2 disaggregates the results by tow type. The contrast is striking:

- **Poverty tows:** The policy caused a large and highly significant increase of 6.1 percentage points in redemption rates.
- **Non-poverty tows:** The policy effect was much smaller, at about 1.0 percentage points, though still statistically significant.

These findings demonstrate that the waiver's impact was overwhelmingly concentrated on tows linked to financial hardship. Additional analyses, including event studies (Appendix B) and robustness checks using both the 2020 and 2021 policy estimates (Appendix C), further support this interpretation.

Although the June 2021 rule change shows no consistent or broad-based effect, one subgroup in Table C.4—vehicles more than 20 years old—exhibits a statistically significant 5.7 percentage-point increase in redemption rates. We interpret this cautiously: because the 2021 reform limited which vehicles could be towed for debt-related reasons, it likely changed the composition of vehicles entering the tow pool rather than drivers' redemption behavior. This selection effect may explain the apparent increase among the oldest vehicles.

## 5 Discussion

### 5.1 Potential mechanisms

The primary mechanism driving the observed increase in vehicle redemption rates is direct financial relief. The August 2020 policy drastically reduced the cost of retrieval for low-income individuals,

moving the total fee from a prohibitive level (often over $500) to a more manageable one (around $100). For an individual facing financial insecurity, this reduction fundamentally alters the economic calculation of whether to reclaim a vehicle. This mechanism is strongly supported by the heterogeneity analysis, which found that the policy's effect was largest for the owners of the oldest vehicles. As vehicle age is inversely correlated with market value, the owners of these cars are the most sensitive to high retrieval costs, and thus benefited most from the fee reduction.

A secondary, more complex mechanism relates to the compositional effects of the June 2021 rule change. By preventing some of the most financially distressed individuals (i.e., verified homeless persons) from being towed for debt-related reasons, this policy likely altered the composition of the remaining towed vehicle population. However, our analysis finds no significant additional impact on redemption rates from this policy, suggesting that the direct financial relief of the 2020 waiver is the dominant mechanism explaining the change in outcomes for vehicles that are towed.

## 5.2 Limitations

While the quasi-experimental design provides a strong basis for causal inference, this study has several limitations. First, the analysis relies on a proxy for owner income based on vehicle age and make. While this is a standard practice in transportation research where direct income data is unavailable, the proxy is imperfect. Misclassification of some owners into the wrong group would likely lead to attenuation bias, meaning our estimates should be interpreted as a conservative lower bound.

Second, the study period overlaps with the unprecedented shocks of the COVID-19 pandemic. The suspension of enforcement, phased reintroduction, shifts in travel behavior, and disbursement of federal stimulus payments all occurred within our analysis window. While the difference-in-differences design helps control for common shocks, these events may have had differential effects on low-income individuals that are not fully captured by the model.

Finally, our event study analysis revealed violations of the parallel trends assumption (Appendix B). For "non-poverty tows," a significant positive pre-trend renders causal claims for this subgroup unreliable. For "poverty tows," a negative pre-trend indicates a persistent disadvantage for low-income groups prior to the intervention. While we argue this strengthens the interpretation that the policy reduced this trajectory, it deviates from the ideal parallel trend. Thus, while the evidence points to a positive causal effect, especially for poverty-related tows, the precise magnitude should be interpreted with caution.

## 6 Conclusion and policy implications

This analysis provides robust evidence that San Francisco's August 2020 tow fee waiver was a highly effective and precisely targeted intervention. A difference-in-differences analysis shows the policy caused a 3.5 percentage point increase in the vehicle redemption rate for low-income owners. This effect was concentrated on "poverty tows"—those for unpaid citations or expired registration—where redemption rates rose by a substantial 6.1 percentage points.

The policy's benefits were greatest for the most vulnerable populations, having nearly double the impact on owners of vehicles 16 years or older compared to those with newer cars. Descriptive evidence further supports this, showing that after the policy, the average age of a successfully redeemed low-income vehicle increased to 17 years, while the average age of cars ultimately abandoned rose to nearly 21 years. As a robustness check, we also analyze the June 2021 procedural reform, which altered towing conditions but left fees unchanged. Consistent with expectations, this rule change had no measurable impact on redemption rates, reinforcing the conclusion that the financial waiver was the central driver of improved outcomes.

These findings have clear policy implications. They demonstrate that direct financial relief is a powerful tool for preventing asset forfeiture among low-income populations, providing a data-driven case for maintaining and expanding such programs. The concentration of the policy's effect on "poverty tows" aligns with the growing legal and advocacy consensus that using towing as a debt-collection mechanism is an inequitable practice. This research provides empirical evidence that financial interventions can effectively mitigate these harms and prevent a parking violation from becoming a financial crisis for vulnerable residents.

## AI assistance statement

We used Google's Gemini large language model (LLM) to assist with proofreading, phrasing improvements, code debugging, and analysis during manuscript preparation. No custom orchestration, fine-tuning, or tool integrations were used.

## Broader impact

This research provides empirical evidence that direct financial relief policies can significantly improve vehicle redemption outcomes for low-income populations. By demonstrating that San Francisco's 2020 fee waiver program reduced the disproportionate burden of "poverty tows," our study contributes to ongoing policy discussions around equity in municipal enforcement practices. The broader societal impact lies in showing how targeted interventions can prevent asset loss and economic destabilization for vulnerable households.

This paper was conducted in collaboration with an AI system that contributed at every stage, including literature review, methodological design, statistical analysis, and writing. In line with the conference's vision of AI scientists as coauthors, we ensured that AI contributions were directed, audited, and critically evaluated by human researchers. The use of AI allowed for a more comprehensive synthesis of existing research and improved clarity of exposition, but human oversight remained central to safeguarding research integrity, ethical standards, and responsible deployment of the findings.

## Reproducibility statement

We provide a detailed description of the empirical strategy, including model specifications, controls, and fixed effects. All regression equations and variable definitions are fully documented in the paper and the appendix. While the administrative towing dataset cannot be publicly released due to privacy restrictions, the methods are transparent and can be applied to similar datasets in other jurisdictions. Code implementing the difference-in-differences and triple-difference analyses can be shared with qualified researchers upon request, ensuring that the main results are reproducible. Potential sources of bias and violations of assumptions are also explicitly discussed to allow readers to assess the robustness of our findings.

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

# Appendix

## A    Key variable construction

This discussion discusses the definition of key variables in more detail. Table A.1 describes the logic used to create the low-income proxy variable based on 2 vehicle characteristics: age and vehicle make. Table A.2 describes how towing incidents are classified as "poverty tows", based on a set of tow reasons listed in the table.

Table A.1: Definition of Low-Income Proxy Variable

| |
|---|
| **Logic:** A vehicle is classified into the low-income proxy group if it meets two criteria: |
| 1. The vehicle's age is greater than 10 years at the time of the tow. |
| 2. The vehicle's make is not on the pre-defined list of luxury makes. |
| **Luxury Makes:** ACURA, ALFA ROMEO, ASTON MARTIN, AUDI, BENTLEY, BMW, FERRARI, INFINITI, JAGUAR, LAMBORGHINI, LAND ROVER, LEXUS, MASERATI, MERCEDES, MINI, PORSCHE, ROLLS ROYCE, ROVER, SAAB, TESLA, VOLVO |

Notes: This proxy construction is a standard and necessary practice in transportation and economic research where direct income data is unavailable in administrative datasets. The use of vehicle age, value, and make as a proxy for owner socioeconomic status is a well-established methodology.

Table A.2: Classification of Poverty Tows

| Category | Tow Reason Description |
|---|---|
| **Unpaid Citations** | 'SCOF/651.I - Scofflaw - Citations' 
 'CITATIONS - Citations and Boot Fees' 
 'SCOF/STOP-SWP - STOP Sweep Scofflaw' |
| **Expired Registration** | 'SCOF/651.O - Scofflaw - Expired Registration' 
 'SCOF/4000A - Scofflaw - Expired Registration - Moving' 
 'SCOF/5204A - Expired Registration' |
| **Parking Over 72 Hours** | '37A - Parked More than 72 Hours' 
 'TRC7.2.29 - PARKING OVER 72HR' |
| **Combined** | 'SCOF/651.I-O - Scofflaw - Citations - Expired Registration' |

Notes: This classification is based on the definition of "poverty tows" used by legal advocates and in legislative analyses, which identifies tows for non-safety, administrative violations that are strongly correlated with financial poverty.

## B    Event study

This appendix presents the detailed results from our event study analyses, which provide a dynamic, month-by-month visualization of the policies' impacts. The primary purpose of an event study is to test the validity of the core identifying assumption of the difference-in-differences (DiD) framework: the parallel trends assumption. This assumption requires that, in the absence of the policy intervention, the redemption rates for the low-income and control groups would have followed similar trends. By plotting the coefficients for the months preceding the policy, we can visually inspect whether this assumption holds.

We present three separate event study analyses, each using our fully saturated model specification with a +/- 24-month window around the August 2020 fee waiver. We begin with an event study

on the full sample of tows. As the results will show, this analysis reveals a significant violation of the parallel trends assumption, making a causal interpretation on the full sample problematic. This finding motivates the subsequent split-sample analysis, which separates tows into "poverty" and "non-poverty" categories. This allows us to investigate the source of the pre-trend violation and to isolate a subgroup for which a causal claim may be more methodologically sound.

The following plots provide crucial context for the DiD results presented in the main text. The event study for "non-poverty tows" serves as a powerful placebo test, visually confirming a severe pre-trend that invalidates causal claims for this subgroup. In contrast, the event study for "poverty tows" reveals a different pre-policy dynamic—not a converging trend, but a persistent, negative gap that establishes the baseline disparity the policy was designed to address. While the month-to-month estimates in these plots can be noisy due to statistical power limitations, they illustrate the dynamic gaps that are averaged in our more powerful DiD models. The DiD estimate of a +6.1 percentage point effect for poverty tows should be interpreted as the average amount by which the 2020 fee waiver successfully closed this pre-existing negative gap.

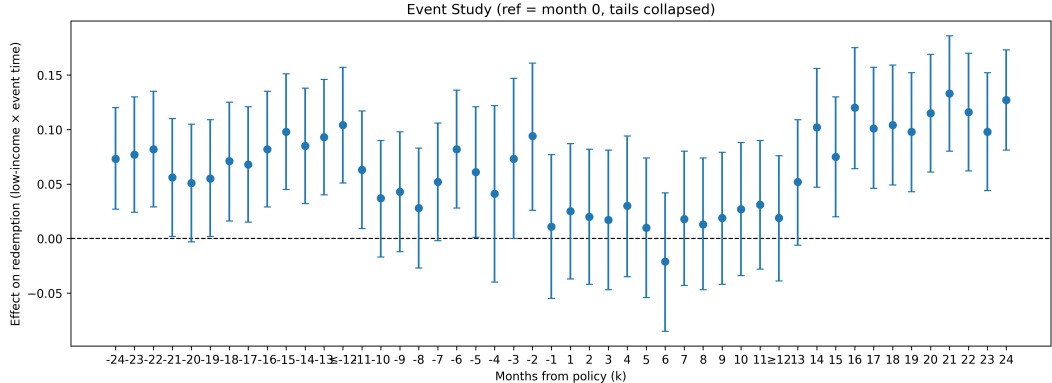

Figure B.1: Event Study Analysis

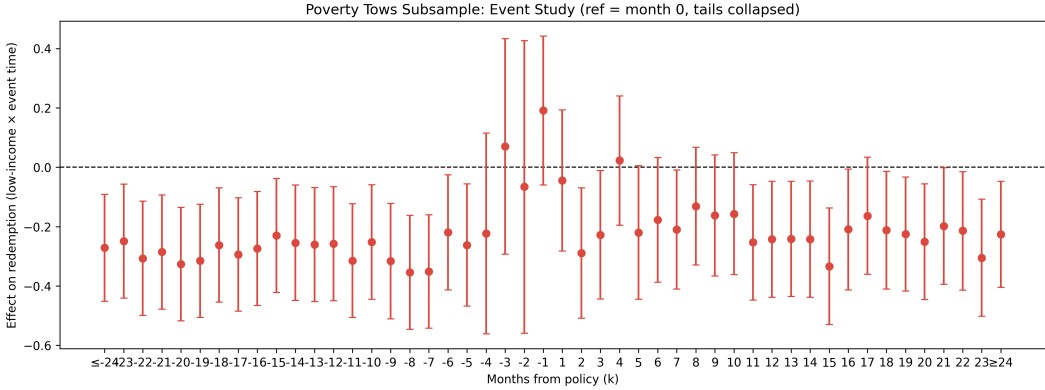

Figure B.2: Poverty Tows Subsample Event Study Analysis

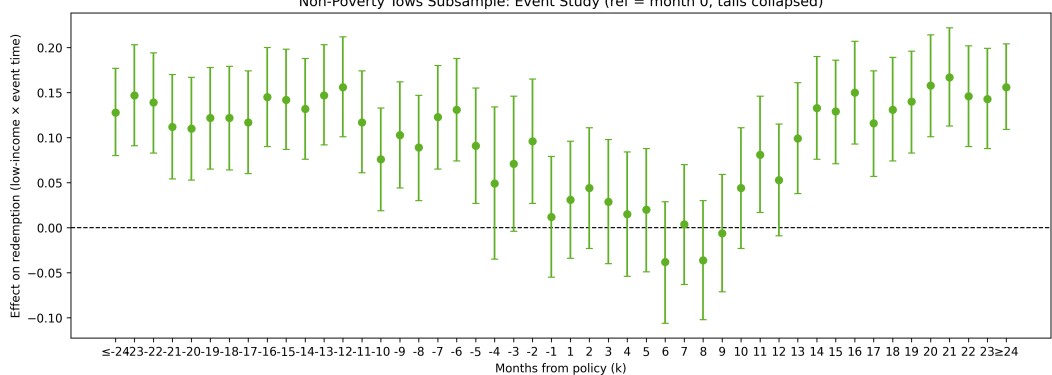

Figure B.3: Non-Poverty Tows Subsample Event Study Analysis

# C Main specification and heterogeneity analyses

This appendix provides the detailed regression output corresponding to the coefficient plots and analyses presented in the main body of the paper. Table C.3 presents the full results from the four specifications of the main difference-in-differences model, demonstrating the robustness of the primary causal estimate. Table C.4 contains the output from the triple-difference model exploring treatment effect heterogeneity by vehicle age group. Finally, Table C.5 shows the results from the difference-in-differences models run on the separate subsamples of poverty and non-poverty tows.

Table C.3: Difference-in-Differences: Robustness of the 2020 Fee Waiver Effect

|  | (1) Raw DiD | (2) Controls | (3) Fixed Effects | (4) Full Model |
|---|---|---|---|---|
| Post_Waiver2020 × Low-Income | 0.039*** | 0.031*** | 0.045*** | 0.035*** |
|  | (0.004) | (0.004) | (0.004) | (0.004) |
| Post_Waiver2021 × Low-Income | -0.005 | 0.004 | -0.005 | 0.005 |
|  | (0.007) | (0.007) | (0.007) | (0.007) |
| Controls | No | Yes | No | Yes |
| Time Fixed Effects | No | No | Yes | Yes |
| Observations | 269,611 | 269,611 | 269,611 | 269,611 |

Notes: The dependent variable is a binary indicator for whether a vehicle was redeemed. The table displays the interaction coefficients from four different specifications of the difference-in-differences model. 'Post_Waiver2020' is an indicator for the period after August 1, 2020. 'Low-Income' is the low-income proxy indicator. Controls include indicators for hardship tow, repeat tow, and out-of-state registration. Time Fixed Effects include month-year and day-of-month fixed effects. Robust standard errors are in parentheses. $^{***}p < 0.01$, $^{**}p < 0.05$, $^{*}p < 0.1$.

Table C.4: Triple-Difference: Fee Waiver Effect by Vehicle Age Group

|  | Redeemed |
| --- | --- |
| *Effect of August 2020 Fee Waiver* |  |
| Post2020 × Low-Inc 10–15 yrs | 0.029*** |
|  | (0.005) |
| Post2020 × Low-Inc 16–20 yrs | 0.073*** |
|  | (0.006) |
| Post2020 × Low-Inc >20 yrs | 0.058*** |
|  | (0.007) |
|  |  |
| *Additional Effect of June 2021 Rule Change* |  |
| Post2021 × Low-Inc 10–15 yrs | -0.009 |
|  | (0.008) |
| Post2021 × Low-Inc 16–20 yrs | 0.007 |
|  | (0.011) |
| Post2021 × Low-Inc >20 yrs | 0.057*** |
|  | (0.014) |
| Observations | 269,611 |

Notes: The dependent variable is a binary indicator for whether a vehicle was redeemed. The table displays interaction coefficients from the fully saturated triple-difference model. The baseline group for the low-income interactions is the control group (all non-low-income vehicles). All regressions include controls for hardship tow, repeat tow, out-of-state registration, and day-of-week, month-year, and day-of-month fixed effects. Robust standard errors are in parentheses. $^{***}p < 0.01$, $^{**}p < 0.05$, $^{*}p < 0.1$.

Table C.5: Difference-in-Differences: Fee Waiver Effect by Tow Type

|  | (1) Poverty Tows | (2) Non-Poverty Tows |
| --- | --- | --- |
| Post_Waiver2020 × Low-Income | 0.061*** | 0.009** |
|  | (0.010) | (0.004) |
|  |  |  |
| Post_Waiver2021 × Low-Income | -0.018 | 0.004 |
|  | (0.017) | (0.007) |
| Observations | 68,222 | 201,389 |

Notes: The dependent variable is a binary indicator for whether a vehicle was redeemed. The table displays the interaction coefficients from the fully saturated difference-in-differences model, run on two separate subsamples. All regressions include controls for repeat tow, out-of-state registration, and day-of-week, month-year, and day-of-month fixed effects. Robust standard errors are in parentheses. $^{***}p < 0.01$, $^{**}p < 0.05$, $^{*}p < 0.1$.

# D Compositional analysis

To complement the causal analysis, we conduct a compositional analysis to provide descriptive evidence on how the profile of successfully redeemed vehicles changed after the policy intervention. We compare the characteristics of vehicles reclaimed from the impound lot in the period before the August 2020 fee waiver to those reclaimed in the period after. Specifically, we examine the mean and median vehicle age and the market share of the most common vehicle makes. This comparison

is performed on two distinct populations: first, the full sample of all redeemed vehicles, and second, a restricted subsample of redeemed vehicles belonging to our low-income proxy group. This descriptive exercise allows us to observe whether the policy altered the types of cars being saved from abandonment, offering a clearer picture of who benefited most from the financial relief.

Table D.6: Compositional Analysis of Redeemed Vehicles

| | All Redeemed Vehicles | | Redeemed Low-Income Proxy | |
| --- | --- | --- | --- | --- |
| | Pre-Policy | Post-Policy | Pre-Policy | Post-Policy |
| *Vehicle Age* | | | | |
| Mean | 8.32 | 9.98 | 15.93 | 17.01 |
| Median | 6.00 | 8.00 | 15.00 | 16.00 |
| | | | | |
| *Top 5 Makes (% Share)* | | | | |
| TOYOTA | 18.2% | 19.4% | 24.8% | 26.6% |
| HONDA | 13.9% | 13.9% | 22.9% | 20.4% |
| FORD | 8.6% | 7.7% | 10.3% | 11.1% |
| NISSAN | 6.5% | 5.4% | 5.7% | 6.0% |
| CHEVROLET | 5.2% | 5.0% | 6.6% | 6.8% |
| Observations (N) | 69,057 | 109,158 | 20,452 | 37,773 |


